Bagworm bags as portable armour against invertebrate predators

Sugiura Shinji sugiura.shinji@gmail.com
Graduate School of Agricultural Science, Kobe University , Kobe , Japan
Bertram Susan
Electronic publication date: 2016 Feb 15
Publication date: 2016
Volume: 4
Electronic Location ID: e1686
Received 2015 Aug 18; Accepted 2016 Jan 22
Copyright: ©2016 Sugiura
Copyright year: 2016
Copyright holder: Sugiura
License: This is an open access article distributed under the terms of the Creative Commons Attribution License, which permits unrestricted use, distribution, reproduction and adaptation in any medium and for any purpose provided that it is properly attributed. For attribution, the original author(s), title, publication source (PeerJ) and either DOI or URL of the article must be cited.
License URL: https://creativecommons.org/licenses/by/4.0/

Keywords: Carabidae, Physical defense, Portable cases, Predation, Psychidae

Funding: Grant-in-Aid for Scientific Research 25292034 This research was partly supported by a Grant-in-Aid for Scientific Research (no. 25292034). The funders had no role in study design, data collection and analysis, decision to publish, or preparation of the manuscript.

==============================
Some animals have evolved the use of environmental materials as “portable armour” against natural enemies. Portable bags that bagworm larvae (Lepidoptera: Psychidae) construct using their own silk and plant parts are generally believed to play an important role as a physical barrier against natural enemies. However, no experimental studies have tested the importance of bags as portable armour against predators. To clarify the defensive function, I studied the bagworm Eumeta minuscula and a potential predator Calosoma maximoviczi (Coleoptera: Carabidae). Under laboratory conditions, all bagworm larvae were attacked by carabid adults, but successfully defended themselves against the predators’ mandibles using their own bags. The portable bags, which are composed mainly of host plant twigs, may function as a physical barrier against predator mandibles. To test this hypothesis, I removed the twig bags and replaced some with herb leaf bags; all bag-removed larvae were easily caught and predated by carabids, while all bag-replaced larvae could successfully defend themselves against carabid attacks. Therefore, various types of portable bags can protect bagworm larvae from carabid attacks. This is the first study to test the defensive function of bagworm portable bags against invertebrate predators.

Introduction

Animals have evolved defensive armour to protect themselves from predators; for example, armadillos and crabs have hardened their exoskeletons, hedgehogs and sticklebacks have developed spines, and snails have developed shells as defensive armour (Edmunds, 1974; Eisner, 2003; Emlen, 2014). Conversely, many animals have evolved the use of environmental materials as defensive armour (Edmunds, 1974). For example, phytophagous insects accumulate host plant secondary metabolites in their bodies to defend themselves chemically against their natural enemies (e.g., Eisner, Esiner & Siegler, 2005), and hermit crabs use gastropod shells as “portable armour” against predators (Edmunds, 1974).

The larvae of holometabolous insects are vulnerable to enemy attacks because of their soft bodies, and have developed various types of defensive armour (Greeney, Dyer & Smilanich, 2012). For example, the spines and hairs of caterpillars constitute physical defences against predators (Dyer, 1995; Dyer, 1997; Murphy et al., 2009; Sugiura & Yamazaki, 2014). Some insect larvae construct “portable cases” using their own silk thread, excrement, and/or environmental materials (e.g., plant parts and stones). Such case-bearing behaviour has been found in three holometabolous insect orders (Root & Messina, 1983): Trichoptera (e.g., caddisfly larvae of the suborder Integripalpia; Holzenthal et al., 2007); Coleoptera (e.g., leaf beetle larvae of the subfamilies, Clytrinae, Cryptocephalinae, Chlamisinae, and Lamprosomatinae; Brown & Funk, 2005; Chaboo, Brown & Funk, 2008); and Lepidoptera (e.g., moth larvae of the superfamilies Incurvarioidea and Tineoidea; Stehr, 1987). Physical defence against predators using portable cases has been tested experimentally in Trichoptera (Otto & Svensson, 1980; Ferry et al., 2013) and Coleoptera (Root & Messina, 1983; Brown & Funk, 2010), but not in Lepidoptera.

The bagworm family Psychidae (Lepidoptera: Tineoidea) includes ca. 1000 species, and all of their larvae construct portable cases (Rhainds, Davis & Price, 2009). The materials used for constructing bags differ among bagworm species; e.g., tree/herb/grass leaves, lichens, twigs, petioles, bark fragments, wood debris, and sand particles (Sugimoto, 2009a; Sugimoto, 2009b). The portable bags are generally believed to play an important role as portable armour against natural enemies (Rhainds, Davis & Price, 2009). For example, bags have been reported to function as a physical barrier against parasitoid attack; the ovipositor of an ichneumonid parasitoid was too short to reach pupae of the bagworm Thyridopteryx ephemeraeformis (Haworth) inside the larger bags, and the parasitism rate was inversely correlated with bag size (Cronin & Gill, 1989). However, bagworm larvae and pupae inside bags are generally known to suffer heavier parasitism by more diverse parasitoids than are other external-feeding caterpillars (Hawkins, 1994), suggesting that bagworm bags may not be effective armour against parasitoids. Rather, predators such as birds and predacious arthropods may impose a selective pressure on the evolution or maintenance of bags. Although the impacts of predators have been reported in some bagworm species (Rhainds, Davis & Price, 2009; Pierre & Idris, 2013), no experimental studies have tested the importance of bags and materials used for bags as defensive armour against predators. Clarifying the defensive function of bags would contribute to further understanding of how portable armour has evolved in animals.

To test whether portable bags can protect bagworms from predator attacks, Calosoma adults (Coleoptera: Carabidae) were observed attacking larvae of a bagworm species under laboratory conditions. Adults of the carabid genus Calosoma hunt lepidopteran larvae and pupae (Forsythe, 1982; Weseloh, 1985; Bruschi, 2013), providing a good model predator for investigating the defensive behaviour of lepidopteran larvae (Sugiura & Yamazaki, 2014). In this study, I first investigated the defensive success or failure of bagworm larvae against carabid attacks. Second, I tested whether bag-removed larvae could defend themselves against carabids in order to clarify the importance of bags. Furthermore, I investigated the effects of bag replacement (with a different type of bag) on the defensive success of bagworm larvae against carabid attacks to elucidate the importance of materials for constructing bags.

Materials and Methods

Study species

To clarify the defensive function of portable bags, I used the bagworm species Eumeta minuscula Butler (Psychidae) and the potential predator Calosoma maximoviczi Morawitz (Carabidae).

Larvae of E. minuscula feed on leaves of various woody species, including both angiosperms and gymnosperms, and construct portable bags using their own silk thread and leaf fragments/petioles/twigs of host plants (Figs. 1A and 1B; Kobayashi & Taketani, 1993; Sugimoto, 2009b). In Japan, E. minuscula overwinters as middle-instar larvae and pupates in early summer (Kobayashi & Taketani, 1993). Various natural enemies are known to attack E. minuscula larvae and pupae inside the bags (Kobayashi & Taketani, 1993), including 25 parasitoid wasp species (Nishida, 1983), three parasitoid fly species (Shima, 1999), one ant species (Nishida, 1983), and one bird species (Ikeda, 1988). For laboratory experiments, all E. minuscula larvae were collected from the forest edge in Shimosasori, Takarazuka, Hyogo (34°55′N, 135°18′E, 190 m above sea level) in late May 2015. Active larvae were used in laboratory experiments, although unhatched eggs of parasitoid flies (Diptera: Tachinidae) were found on some active larvae. Before the experiments, I measured the fresh weight of each E. minuscula larva and its bag to the nearest 0.1 mg using an electronic balance (PA64JP, Ohaus, Tokyo, Japan). I also used slide callipers to measure the bag length, larval length, and head capsule width of E. minuscula to the closest 0.1 mm. Sampled larvae were determined to be 6th or 7th (last) instar based on the head capsule width (range: 2.5–3.8 mm; cf. Nishida, 1983). The bags were ca. 1.8 times the length of the larvae (Fig. 1B; mean larval body length, 17.7 ± 2.5 mm (mean ± SD), mean bag length, 32.4 ± 4.9 mm, n = 36) and as heavy as larvae (mean fresh larval weight, 245.3 ± 71.1 mg, mean fresh bag weight, 245.0 ± 74.4 mg, n = 36). All bags were composed mainly of plant twigs (Figs. 1A and 1B).

Figure 1 Photos of the bagworm Eumeta minuscula and its potential predator Calosoma maximoviczi.

(A) Eumeta minuscula bags on shrubs. (B) An E. minuscula larva and the inside of its bag. (C) A bagworm and a carabid on bamboo material under laboratory conditions. (D) A bag protecting the larva from a carabid attack. (E) A bag-removed larva eaten by a carabid. (F) A replaced bag protecting the larva from a carabid attack.

Calosoma maximoviczi adults exclusively hunt lepidopteran larvae on both the ground and vegetation (Kamata & Igarashi, 1995; Sugiura & Yamazaki, 2014). This carabid species uses its mandibles to catch and injure caterpillars, and then feeds on them (Sugiura & Yamazaki, 2014). Since C. maximoviczi adults can attack caterpillars of various species and size under laboratory conditions, C. maximoviczi adults are considered appropriate for investigating the defence behaviour of lepidopteran larvae against generalist predators (Sugiura & Yamazaki, 2014). For laboratory experiments, all adults of C. maximoviczi were collected from a secondary forest in Nunobiki, Kobe, Hyogo (34°42′N, 134°11′E, 60–170 m above sea level), in early May 2015. I have not observed C. maximoviczi adults attacking bagworms under field conditions; however, the habitat and active season partly overlap between E. minuscula larvae and C. maximoviczi adults in this sampling region, suggesting that E. minuscula larvae can encounter C. maximoviczi adults on trunks or twigs of woody plants. Active adults of C. maximoviczi, which attacked caterpillars under laboratory conditions, were used in the laboratory experiments. Before the experiments, I measured the fresh weight of each adult of C. maximoviczi to the nearest 0.1 mg using an electronic balance. I also used slide callipers to measure the body and mandible lengths of C. maximoviczi to the closest 0.1 mm.

The insects used in this study were not endangered or protected species in the sampling region. The experiments were undertaken according to the Kobe University Animal Experimentation Regulations. The experiments also comply with the current laws of Japan.

Laboratory experiments

To test the defensive function of portable bags, I conducted the following experiment in a well-lit laboratory (25 °C) in late May 2015. A bagworm larva and a carabid adult were placed on bamboo material (width 7 mm, height 15 mm; Figs. 1C and 2), which modelled tree twigs and trunks, because both carabids and bagworms forage on tree twigs and trunks under field conditions. The bamboo material was looped (Fig. 2; length 700 mm, diameter 200 mm) so that bagworms could encounter carabids in all trials. The looped bamboo material was also surrounded by a plastic circular cylinder (diameter 220 mm, height 120 mm).

Figure 2 The arena used in the experiments.

A Eumeta minuscula larva and a Calosoma maximoviczi adult were placed on bamboo material.

During a 10-min period, I observed (1) whether a carabid attacked a bagworm larva and (2) whether the carabid finally injured the bagworm. I deemed that a bagworm larva could not defend itself against a carabid adult when the carabid was observed to catch and injure the larva within the 10-min period. When an adult carabid gave up attacking a bagworm without injuring it, I deemed that the bagworm successfully defended itself against the carabid. I also continued observing further attacks by the carabid within the 10-min period. I used 15 adults (4 females and 11 males) of C. maximoviczi (mean ± SD body weight, 447.1 ± 104.7 mg, mean body length, 25.1 ± 2.0 mm, mean mandible length, 1.9 ± 0.1 mm, n = 15) to conduct three types of experiments (Table 1); body weight and length significantly differed among three types of experiments (one-way analyses of variance; body weight, F = 4.3, P = 0.04; body length, F = 4.1, P = 0.04), while mandible length did not differ (F = 0.9, P = 0.44).

Table 1 Defensive success or failure of the bagworm Eumeta minuscula against the potential predator Calosoma maximoviczi under laboratory conditions.

Predator (C. maximowiczi)	First prey (E. minuscula)	Second prey (E. minuscula)a	
No.b	Sex	Weight (mg)	No. b	Bag treatmentc	Weight (mg)d	Defencee	Numbers of attacksf	No.b	Bag treatmentc	Weight (mg)d	Defencee	Numbers of attacksf	
Experiment 1	
C1	Male	384.4	E1	Control	711.3	Success	1	E1	Removed	263.5	Failure	1	
C2	Male	426.3	E2	Control	436.6	Success	4	E2	Removed	226.5	Failure	1	
C3	Male	343.3	E3	Control	485.3	Success	3	E3	Removed	254.1	Failure	1	
C4	Female	530.4	E4	Control	285.8	Success	4	E4	Removed	138.1	Failure	1	
C5	Male	604.2	E5	Control	699.2	Success	1	E5	Removed	326.7	Failure	1	
Experiment 2	
C6	Female	420.9	E6	Removed	338.2	Failure	1	E11	Control	236.8	Success	3	
C7	Male	572.1	E7	Removed	218.5	Failure	1	E12	Control	505.7	Success	1	
C8	Male	600.4	E8	Removed	238.4	Failure	1	E13	Control	330.5	Success	1	
C9	Male	446.3	E9	Removed	118.3	Failure	1	E14	Control	618.0	Success	1	
C10	Female	566.6	E10	Removed	164.7	Failure	1	E15	Control	377.1	Success	1	
Experiment 3	
C11	Male	369.0	E16	Replaced	427.3	Success	2	E16	Removed	230.4	Failure	1	
C12	Male	418.4	E17	Replaced	479.4	Success	3	E17	Removed	359.0	Failure	1	
C13	Female	372.9	E18	Replaced	340.9	Success	3	E18	Removed	205.4	Failure	1	
C14	Male	399.0	E19	Replaced	449.6	Success	1	E19	Removed	336.8	Failure	1	
C15	Male	252.7	E20	Replaced	313.5	Success	4	E20	Removed	202.8	Failure	1	
Notes.

a The second prey was provided to each predator after my observation of the predator behaviour in response to the first prey.

b Different code numbers showed that different individuals were used.

c Bag treatment: control, normal bags; removed, bags were removed experimentally; replaced, normal (twig) bags were replaced with soft (herb leaf) bags (see text).

d Total fresh weight (including bags) was shown for control and bag-replaced larvae, while fresh body weight (except bags) was measured for bag-removed larvae.

e Defence success and failure of E. minuscula indicated predation failure and success by C. maximoviczi, respectively.

f Total number of attacks by C. maximoviczi on E. minuscula. Two or more attacks indicated that a carabid attacked a bagworm again after giving up its first attack.

Experiment 1: Normal E. minuscula larvae (bag treatment, control) were provided as the first prey to five adult C. maximoviczi (Table 1). To clarify the importance of bags as a defensive barrier against predators, I provided bag-removed E. minuscula larvae (bag treatment, removed) as the second prey to the same five carabid individuals just after the experiment with the first prey (Table 1). I used a pair of scissors to remove the bags from E. minuscula larvae; first prey that successfully defended itself against carabid attacks was also used as the second prey. To investigate which bagworm individuals or bag treatments could affect the defence success of bagworms, I used the same bagworms in different treatments. The removal of bags from middle- and late-instar bagworms has been known to render bagworms susceptible to drought and starvation; e.g., bag-removed larvae died after several days (Kaufmann, 1968). However, my preliminary observations showed that bag-removed E. minuscula did not die within the 10-min period due to drought and starvation.

Experiment 2: Bag-removed E. minuscula larvae were provided as the first prey to five adult C. maximoviczi (Table 1). I provided control E. minuscula larvae as the second prey to the same five carabids just after the experiment with the first prey (Table 1). Different E. minuscula larvae were used as the second prey. I conducted this experiment to avoid any potential systematic effects of the first prey on responses to the second prey by carabids.

Experiment 3: Bag-replaced E. minuscula larvae (bag treatment, replaced) were provided as the first prey to five adult C. maximoviczi (Table 1). To clarify the importance of materials for constructing bags, I replaced the normal (tight) bags with soft bags. I used a pair of scissors to remove the bags from 10 E. minuscula larvae. The larvae were placed individually in plastic Petri dishes (90 mm diameter, 30 mm high) with minced leaves of the herb species Artemisia indica var. maximowiczii (Asteraceae). I used a pair of scissors to mince the leaves (mean fragment length, 4.4 ± 1.7 mm, n = 27). Five of 10 E. minuscula larvae constructed sufficiently large bags (bag length > 25 mm) using their own silk thread and the leaf fragments (Fig. 3) one day after placement. The replaced bags were ca. 1.5 times the length of the larvae (mean larval body length, 18.6 ± 2.7 mm, mean bag length, 27.9 ± 1.2 mm, n = 5) and half as heavy as larvae (mean fresh larval weight, 266.9 ± 75.1 mg, mean fresh bag weight, 135.3 ± 35.8 mg, n = 5). Such replacement with a different type of bag has been conducted in another bagworm species (Kaufmann, 1968). Five larvae constructing new bags were used as the first prey in this experiment. Just after conducting the experiment with the first prey (i.e., bag-replaced larvae), I provided bag-removed E. minuscula larvae as the second prey to the same five carabids (Table 1). The first prey that had successfully defended itself against carabid attacks was also used as second prey. To investigate which bagworm individuals or bag treatments could affect the defence success of bagworms, I used the same bagworms in different treatments.

All adult carabids attacked each bagworm within the 10-min period. Even when bagworms did not actively walk, carabids were observed to attack and bite motionless bags. Larval weight, bag weight, total (larval + bag) weight, bag length, and larval length of the E. minuscula used in this study did not differ among the three experiments (one-way analyses of variance; F = 0.2–1.1, P = 0.38–0.84).

Fisher’s exact tests were used to compare the success rate of defence by bagworms between control, bag-removed, and bag-replaced treatments. Considering the independence of the data, I excluded the data for the second prey from the analysis. All analyses were performed using R ver. 2.15.1 (R Development Core Team, 2012).

Results

Experiment 1: all control E. minuscula larvae (n = 5) were attacked by C. maximoviczi adults, but successfully defended themselves against the predator attacks (Table 1; Figs. 1C and 1D). When bagworm larvae were attacked by carabids, the larvae quickly retracted their heads and thoraxes into their bags to escape from the attacks (Fig. 1D; Movie S1). Carabids frequently bit the bags, but could not injure the larvae due to the bag protection (Fig. 1D). Finally, all of the carabids gave up attacking the larvae. Three of five bagworm larvae were attacked by carabids again within the 10-min period, but successfully defended themselves against further attacks (Table 1). The other (two) bagworms remained retracted after the first carabid attack and were not attacked again (Table 1). All of the bag-removed larvae were easily caught and injured by the same individual carabids (Fig. 1E and Table 1; Movie S1). The dorsal, lateral, or ventral abdomens of larvae were the locations injured by carabid mandibles.

Figure 3 Predation success of the carabid Calosoma maximoviczi and defensive success of the bagworm Eumeta minuscula for different bag treatments (control, bag-removal, and bag-replacement).

Experiment 2: all bag-removed larvae (n = 5) were easily caught and predated by carabids (Table 1). All control larvae (n = 5) were attacked by the same individual carabids, but successfully defended themselves against the attacks due to bag protection (Table 1). One bagworm was attacked by the carabid again within the 10-min period, but successfully defended itself against further attacks (Table 1). Other bagworms remained retracted after the first carabid attack and were not attacked again (Table 1).

Experiment 3: all bag-replaced larvae (n = 5) were attacked by carabids, but successfully defended themselves against the attacks (Fig. 1F and Table 1). Carabids frequently bit the soft bags, but could not injure the larvae due to the bag protection (Movie S1). Four of five bagworms were attacked by carabids again within the 10-min period, but successfully defended themselves against further attacks (Table 1). The other bagworm remained retracted after the first carabid attack and was not attacked again (Table 1). All the bag-removed larvae were easily predated by the same individual carabids (Table 1).

The success rate of bagworm defence differed significantly among bag treatments (Fig. 3); the defensive success rate of control, bag-removed, and bag-replaced larvae was 100%, 0%, and 100%, respectively (Table 1; Fisher’s exact test; control vs. bag-removal, P = 0.0008, control vs. bag-replacement, P = 1.0, bag-removal vs. bag-replacement, P = 0.0008).

Discussion

Portable cases of bagworms are generally believed to play an important role as a physical defence against natural enemies (Rhainds, Davis & Price, 2009); however, no studies have tested their effectiveness experimentally. This study demonstrated that bags could protect E. minuscula larvae from C. maximoviczi attacks (Table 1 and Fig. 3). This is the first study to test the defensive function of portable cases against invertebrate predators in Lepidoptera. Although the bag defence of a single bagworm species was shown in this study, my experiment showed that bags made from two different materials (i.e., twig and herb leaf bags) could effectively defend bagworms against the predator (Table 1 and Fig. 3). Accordingly, bags made of other materials may also function as defensive armour against invertebrate predators, although further studies are needed. Studies have clarified the defensive function of portable cases in the two holometabolous insect orders Trichoptera (Otto & Svensson, 1980; Ferry et al., 2013) and Coleoptera (Root & Messina, 1983; Brown & Funk, 2010). Case-bearing behaviours are considered to have evolved independently in Trichoptera and Lepidoptera (Holzenthal et al., 2007; Malm, Johanson & Wahlberg, 2013), although trichopterans and lepidopterans branched from a common ancestor (Holzenthal et al., 2007). This study clarified the defensive function in the order Lepidoptera, strengthening the hypothesis that case-bearing behaviour has repeatedly evolved for anti-predator defence in insects.

I observed attack–defence behaviour in 30 pairs of the predator C. maximoviczi and the prey E. minuscula (Table 1). However, I excluded the data for the second prey from the Fisher’s exact tests, because the same individuals of C. maximoviczi and E. minuscula were used in different experiments (Table 1). Such data could be analysed using a generalised linear mixed model (GLMM) with a binomial error distribution and a logit link, with defensive success or failure (0 or 1) by bagworms as a binary response, bag treatments as fixed factors, and carabid individuals as a random effect. However, all bagworms successfully defended themselves in at least one treatment group (Table 1), thereby extending parameters to infinity when all values in a category were 0 or 1 (cf. Sugiura & Yamazaki, 2014). Therefore, the GLMM could not be conducted in this study. Although the sample size for Fisher’s exact tests was too small (n = 5∕treatment), the combined data showed robust results; i.e., all control group bagworms (n = 15) could successfully defend against carabid attacks, and all bag-removed larvae (n = 10) failed to defend against the attacks (Fig. 3).

No carabid species have been observed preying on bagworm larvae under field conditions. However, I showed that bagworms could perfectly defend against carabid attacks (Table 1 and Fig. 3). Such perfect defence by bagworms suggests very few chances to observe carabid predation on bagworms under field conditions. Other natural enemies are known to impact bagworms (Ellis et al., 2005; Rhainds, Davis & Price, 2009). For example, birds have been considered to regulate bagworm populations (Horn & Sheppard, 1979). However, birds may not prefer bagworms over non-bagged caterpillars because of the increased handling cost (i.e., time taken to remove bags; cf. Moore & Hanks, 2000). Furthermore, the large bags of bagworms have been observed to prevent parasitoid oviposition (Cronin & Gill, 1989). However, more diverse parasitoid species and higher parasitism rates have been reported for case-bearing caterpillars than bare caterpillars (Hawkins, 1994). In fact, a relatively large number of parasitoid species is known to parasitise the bagworm E. minuscula (Nishida, 1983). This may be related to the “refugia” hypothesis; i.e., caterpillars that are unlikely to be eaten by predators can provide enemy-free space for parasitoids (Gentry & Dyer, 2002; Stireman & Singer, 2003). Therefore, indirect interactions among predators and parasitoids via shared prey may alter selection pressures on bag evolution in bagworms. Studies have used predators from various groups, including ants, bugs, and wasps, to test the effectiveness of caterpillar defences against natural enemies (Dyer, 1995; Dyer, 1997; Murphy et al., 2009); however, I used a single predator species in this study. Many interaction factors such as attack size, strategy, and natural history of the predators may cause the variation in defensive effectiveness in caterpillars. Consequently, bagworm defences against predators other than carabid beetles should be tested to clarify the selective agents leading to the evolution of portable bags.

Bagworm bags may have other functions (Rhainds et al., 2009). For example, bags can provide microclimate conditions that protect immature bagworm from desiccation or that accelerate development (Barbosa, Waldvogel & Breisch, 1983; Smith & Barrows, 1991; Rivers, Antonelli & Yoder, 2002; Rhainds et al., 2009). In addition, constructing bags can magnify their relative size to arthropod predators; e.g., bags were ca. 1.8 times the length of larvae in E. minuscula (Fig. 1B). The size magnification by bag construction can provide protection through increased effectiveness of physical or behavioural defences against arthropod predators because predation by arthropods is generally negatively size-dependent (Remmel, Davison & Tammaru, 2011; Greeney, Dyer & Smilanich, 2012). However, one study indicated that C. maximoviczi eventually attacked various sizes and species of lepidopteran larvae (body weight, 33.3–566.7 mg, body length, 12.6–34.6 mm; Sugiura & Yamazaki, 2014). Furthermore, I observed C. maximoviczi adults attacking large hawk moth larvae under laboratory conditions (body weight, 7288.6–16866.9 mg, body length 84.3–112.7 mm), although they did not successfully prey on the large larvae (Sugiura, unpublished data). Therefore, the different predation rate by C. maximoviczi adults between control and bag-removed larvae (Table 1 and Fig. 3) was not caused by the size difference between control and bag-removed larvae, but by the presence/absence of bags. Furthermore, the cryptic appearance can also serve as camouflage (Rhainds et al., 2009), and although this study did not test the importance of cryptic appearance for bagworms, C. maximoviczi adults were frequently observed to attack and bite motionless bags of E. minuscula larvae. This suggests that carabids can use scent as well as appearance to locate prey. Therefore, the cryptic appearance of bagworms was unlikely to influence my results. Taken together, bagworm bags may have various types of functions that are not mutually exclusive. Portable cases that have more than one function may be selected more frequently and evolve more rapidly than those with a single function.

Supplemental Information

Movie S1 Movie of bagworm and carabid behavior

A movie showing the carabid Calosoma maximoviczi attacking normal, bag-removed, and bag-replaced larvae of Eumeta minuscula under laboratory conditions. Normal and bag-replaced larvae could successfully defend themselves against carabid mandibles due to bag protection, while bag-removed larvae were easily predated by carabids.

Click here for additional data file.

I thank the editor and two reviewers for critical comments on my manuscript.

Additional Information and Declarations

Competing Interests

Author Contributions

Ethics

Data Availability

The author declares there are no competing interests.

Shinji Sugiura conceived and designed the experiments, performed the experiments, analyzed the data, contributed reagents/materials/analysis tools, wrote the paper, prepared figures and/or tables, reviewed drafts of the paper.

The following information was supplied relating to ethical approvals (i.e., approving body and any reference numbers):

The experiments were undertaken according to the Kobe University Animal Experimentation Regulations.

The following information was supplied regarding data availability:

Raw data is shown in Table 1.

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
