# Peer review of "Bagworm bags as portable armour against invertebrate predators"

_PeerJ, doi:10.7717/peerj.1686_

## Round 0.1 · original submission · Major Revisions

Please pay careful attention to Reviewer 2's comments, as I strongly agreed with all of them. In addition to Reviewer 2's suggestions about the statistics, I was particularly confused by your statement about data analysis: lines 186-193. It seems like you attempted to run a GLMM, but then couldn't because all bagworms successfully defended themselves in at least one group. If this is the case (you didn't use this analysis), then why include discussion of it at all? Instead, just discuss the statistics you did use. If you were able to use the GLMM, then you must show those results somewhere.

Please also be sure to address the other important comments from Reviewer 2. These include: secondary matabolite parallels, potential stabilizing selection from predators and parasites and the confusion, predation by birds and how that could be a factor and it is untested in your system, and experimental design issues: lack of standardized use of repeated prey and justification of size.

·

Basic reporting

All bagworm species develop within a self-constructed bag made of silk and plant material. The ecological-physiological functions of the bag cannot be extrapolated from one another, because a bagworm without a bag is .... not a bagworm anymore... removing bagworm larvae from their bag render them not only more susceptible to predation but also to drought, starvation (larvae without a bag don't feed), etc. Trying to make a case that bags evolve to reduce predation is irrelevant because the bag is an inherent, constitutive holistic trait that define bagworms as they are

Experimental design

N = 5 for all treatments ... too low to have any ecological significance

Validity of the findings

trivial finding that does not deserve publication

·

Basic reporting

Bagworm bags as portable armour against invertebrate predators (Sugiura)
The author presents a short but novel experiment to test the effectiveness of bagworms’ “bags” on survival under predation from a species of generalist ground beetle. The three complementary experiments offer brief but useful insights into the system, and despite the small sample size the size of the effect (100% survival with bags [or replaced bags] and 100% mortality without) is clear. The video is particularly useful to understand the nature of the interaction. The brevity of the article renders its slight sample sizes less of an issue.

You mention that case-bearing behaviour may have repeatedly evolved (line 235-236). However, Lepidoptera and Trichoptera share a relatively recent common ancestor and together comprise the Amphiesmenoptera. This statement is unreferenced and needs to be supported and more nuanced.

The introduction could be enhanced to include further discussion of anti-predator defences rather than diving straight into the bags. For instance, the accumulation of physical debris from the environment seems analogous to the accumulation of secondary metabolites in aphids. This would be an interesting parallel to draw, as there are many studies on other kinds of acquired, environmental defences.

You mention defense against parasitoids a couple of times, and in a confusing way. This is important, as it may provide balancing selection against the bags. You should be clear about the relationship (or at least provide more discussion of the inconsistencies in the literature).

Minor issues:
1. L68: may not be effective
2. L70: predators has
3. L106-107: do you have the capsule widths for reference?
4. L141: do you know the duration of the trials?
5. L182-184: the description of the ANOVA test does not make sense here. Either don’t include it and (briefly) discuss the statistics before the presentation of the results in lines 143-146 or move the results of the statistics to the results section.
6. L241-242: “…more parasitoid species are known to parasitise Eumeta bagworms…” than what?

Experimental design

There are a couple of (relatively minor) issues with the experimental design, although the results are likely to be robust to those problems. Problems include the removal of the prey as soon as a predator-prey interaction had occurred. Usually predation trials are run for a period of time rather than a number of interactions. You should state the duration of each trial. Could the carabid not have eaten through the bag eventually…? A second issue is the repeated use of the same prey in some experiments. This could have been built into the experiment in a standardised way, but you use different animals for other treatments.

You justify the lack of size-based predation by citing a study of size-breadth of prey (line 257-259) but your bagged worms clearly fall at (and sometimes beyond, given the SD) the upper limit of that range. Also, while potential size range is useful, the probability of attacking different sizes of prey will likely vary.

Validity of the findings

You justify the selection of the predator because the carabid is a generalist carabid. However, avian predation has been implicated in bagworm population regulation (e.g. HORN, D. J. and SHEPPARD, R. F. (1979), Sex ratio, pupal parasitism, and predation in two declining populations of the bagworm, Thyridopteryx ephemeraeformis (Haworth) (Lepidoptera: Psychidae). Ecological Entomology, 4: 259–265.). You should discuss the role for bags in deterring different types of predators and cite evidence of carabid predation on these species. In fact, the only reference I could find of a trophic interaction between carabids and bagworms was of a bagworm eating carabids (Perisceptis carnivora). You need to find the actual invertebrate predators of bagworms to make this clearer, otherwise your title is over-general. There is a useful (though brief) overview of the predators of bagworms here: Ellis JA, Walter AD, Tooker JF et al. (2005) Conservation biological control in urban landscapes: Manipulating parasitoids of bagworm (Lepidoptera: Psychidae) with flowering forbs. Biological Control, 34, 99-107.

---

## Round 0.2 · accepted · Accept

You did an admirable job of dealing with all reviewers comments. Your revision was thorough and I feel it substantially improved your manuscript. I thank you for taking such care in your revisions and in your rebuttal letter.